# iPla2β Deficiency Suppresses Hepatic ER UPR, Fxr, and Phospholipids in Mice Fed with MCD Diet, Resulting in Exacerbated Hepatic Bile Acids and Biliary Cell Proliferation

**DOI:** 10.3390/cells8080879

**Published:** 2019-08-12

**Authors:** Yanan Ming, Xingya Zhu, Sabine Tuma-Kellner, Alexandra Ganzha, Gerhard Liebisch, Hongying Gan-Schreier, Walee Chamulitrat

**Affiliations:** 1Department of Internal Medicine IV, University of Heidelberg Hospital, Im Neuenheimer Feld 410, 69120 Heidelberg, Germany; 2Institute of Clinical Chemistry and Laboratory Medicine, University of Regensburg, Franz-Josef-Strauss-Allee 11, 93053 Regensburg, Germany

**Keywords:** PLA2G6, endoplasmic reticulum, phospholipids, bile acids, lean NASH, unfolded protein response

## Abstract

**Background:** Group VIA calcium-independent phospholipase A2 (iPla2β) regulates homeostasis and remodeling of phospholipids (PL). We previously showed that iPla2β^−/−^ mice fed with a methionine-choline-deficient diet (MCD) exhibited exaggerated liver fibrosis. As iPla2β is located in the endoplasmic reticulum (ER), we investigated the mechanisms for this by focusing on hepatic ER unfolded protein response (UPR), ER PL, and enterohepatic bile acids (BA). **Methods:** Female WT (wild-type) and iPla2β^−/−^ mice were fed with chow or MCD for 5 weeks. PL and BA profiles were measured by liquid chromatography-mass spectrometry. Gene expression analyses were performed. **Results:** MCD feeding of WT mice caused a decrease of ER PL subclasses, which were further decreased by iPla2β deficiency. This deficiency alone or combined with MCD downregulated the expression of liver ER UPR proteins and farnesoid X-activated receptor. The downregulation under MCD was concomitant with an elevation of BA in the liver and peripheral blood and an increase of biliary epithelial cell proliferation measured by cytokeratin 19. **Conclusion:** iPla2β deficiency combined with MCD severely disturbed ER PL composition and caused inactivation of UPR, leading to downregulated Fxr, exacerbated BA, and ductular proliferation. Our study provides insights into iPla2β inactivation for injury susceptibility under normal conditions and liver fibrosis and cholangiopathies during MCD feeding.

## 1. Introduction

Non-alcoholic fatty liver disease (NAFLD) is one of the most common causes of chronic liver disease worldwide, which describes a spectrum of steatosis, non-alcoholic steatohepatitis (NASH) to cirrhosis, which may progress to primary liver cancer. Reports on the prevalence of NAFLD have suggested that 25% of the general population in the USA and 40–90% of the global obese population have this disease [1]. Although obesity is the main risk factor for NAFLD development, it can also develop in lean subjects [2] found in different ethnic Asian populations with a prevalence of 20% in India, 15% in Japan and China, and 12–13% in Greece and South Korea [3,4]. Among lean subjects in the USA, 19% would have NAFLD, and 12% would have NASH [5]. These patients show an increased risk of all-cause and cardiovascular mortality compared to lean subjects [6]. Lean NAFLD is the most frequent cause of cryptogenic liver disease [7]. Factors beyond obesity may play a role in lean NAFLD advances to more severe disease, such as fibrosis [8]. In addition to visceral obesity and high fructose and cholesterol intake, genetic risk factors, such as patatin-like phospholipase domain-containing 3, are associated with lean NAFLD, and these patients show a decrease of blood lysophosphatidylcholine (LPC) [9]. Moreover, insufficiency of hepatic phosphatidylethanolamine N-methyltransferase, which catalyzes the conversion of phosphatidylethanolamine (PE) to phosphatidylcholine (PC), is reported as a risk for lean NASH [10]. An inhibition of hepatic PC synthesis by feeding mice with methionine- and choline-deficient (MCD) diet is also shown to induce lean NASH associated with a significant reduction of serum LPC [11]. Hence, ample data have shown that disturbance of hepatic phospholipid (PL) metabolism is involved in the development of lean NAFLD/NASH.

Group VIA calcium-independent phospholipase A2 (iPla2β or PLA2G6) hydrolyzes PL at the sn-2 position to generate lysoPL and fatty acid, thus regulating homeostasis and remodeling of PL [12]. Genome-wide meta-analysis reveals a significant association between blood lipids and body fat percentage [12,13]. We have shown that iPla2β^−/−^ mice have been protected from obesity and hepatic steatosis in *ob/ob* [14] and chronic high-fat-diet (HFD)-fed [15] mice with a mechanism of PL replenishment and correction of PL remodeling defect. In MCD-induced lean NASH model, iPla2β^−/−^ mice are, however, not protected from hepatic steatosis and, on the other hand, show exaggerated hepatic fibrosis with hepatic stellate cell activation [16]. This indicates an opposing role of iPla2β inactivation between obese NAFLD and lean NASH models. Further investigations are, therefore, needed to gain insights into the possible role of iPla2β inactivation on the severity of MCD-induced NASH, which is considered more inflammatory than NAFLD induced by HFD [17].

Interestingly, it has been shown that hepatic endoplasmic reticulum (ER) stress plays a significant role in the pathogenesis of obese NAFLD [18], but not of MCD-induced lean NASH [19]. This implies that ER stress and events in the ER in obese NAFLD [14,15] and lean NASH [16] may be modulated differently by iPla2β. Furthermore, we have shown that iPla2β deficiency in the intestine elicits a suppressive effect on an ER unfolded protein response (UPR) protein X-box binding protein-1 (Xbp-1) [20]. As iPla2β is localized in the ER [21], it is hypothesized that iPla2β inactivation in MCD-fed mice may have an effect on hepatic ER UPR proteins and alter the composition of ER PL. Here, we showed a severe defect in the remodeling of ER PL and a suppressive effect of not only hepatic ER UPR proteins but also farnesoid X-activated receptor (Fxr). As Fxr is a regulator of hepatic production of bile acids (BA) [22], which are involved in lean NASH [11], obese NAFLD [23], and hepatic fibrosis [24], we, therefore, determined BA contents in the enterohepatic circulation and biliary epithelial cell proliferation. In this report, we provided molecular mechanisms for the effects of iPla2β inactivation during MCD-induced lean NASH [16] involving alterations of ER PL and downregulation of homeostatic genes, which led to an increase of BA and potentially biliary liver disease.

## 2. Methods and Methods

### 2.1. Animals and Feeding

iPla2β^−/−^ mice were kindly provided by Dr. John Turk (Washington University School of Medicine, St. Louis, MO, USA) as global deletion of exon 9 in iPla2β gene, and genotyping was performed according to published work [14,15,16]. The breeding of all mice was performed at the animal facility of the University of Heidelberg. The cohort consisted of 24 female WT (wild-type) and iPla2β^−/−^ mice at ~12 months of age. Female C57BL/6 mice were used as WT controls. Mice were grouped into feeding with chow (catalog# E15654-04, ssniff Spezialdiäten GmbH, Soest, Germany) or MCD diet (catalog# E15653-94, ssniff Spezialdiäten GmbH) for 5 weeks. Mice were starved for 4 h before sacrifice, and blood was collected. Liver and intestine were either fixed in 10% neutral-buffered formalin or snap-frozen and stored at −80 °C. All animal experiments were approved by the Animal Care and Use Committee of the University of Heidelberg.

### 2.2. Isolation of Hepatic ER

Isolation and purification of ER from mouse liver were performed according to a published method [25]. In brief, 300 mg liver was homogenized in 1 mL ice-cold homogenization buffer (0.5 M sucrose, 1% dextran, 37.5 mM Tris, 5 mM MgCl_2_, and 10 μL/mL protease inhibitor cocktails) in a Bullet Blender (Next Advance Inc., Averill Park, NY, USA) using 1-mm zirconium oxide beads. Homogenates were centrifuged at 5000× *g* at 4 °C for 15 min. Supernatants were diluted in homogenization buffer and centrifuged at 8500× *g* at 4 °C for 5 min. Sucrose solutions at 1.3 M, 1.5 M, and 2.0 M were prepared by using 37.5 M Tris buffer, pH 6.4, containing 1% dextran, 5 mM MgCl_2_, 1 mM DTT, and 0.1 mM PMSF. Supernatants were top-loaded over a discontinuous 3-layer sucrose gradient with *v/v/v* of 3:4:4. Gradients were subjected to ultracentrifugation (Beckman Optima XL-90, Beckman Coulter GmbH, Krefeld, Germany) using an SW41 TI rotor at 90,000× *g* for 90 min. ER fractions I and II between 1.3 M–1.5 M and 1.5 M–2.0 M interfaces, respectively, were collected and diluted with buffer, pH 7.4, containing 55 mM Tris, 5 mM MgCl_2_, 1 mM DTT, and 0.1 mM PMSF. The mixture was again centrifuged at 90,000× *g* for 20 min. The ER pellets were resuspended in ice-cold buffer, pH 7.4, containing 0.25 mM sucrose, 10 mM Tris, 1 mM DTT, 0.1 mM PMSF, and 10 μL/mL protease inhibitor cocktails (Calbiochem, Darmstadt, Germany). Protein contents in ER fractions were determined.

### 2.3. Western Blot Analyses

Proteins from liver homogenates (30 μg) or ER fractions (10 μg) were separated by SDS-PAGE and transferred onto a polyvinylidene difluoride membrane. Membranes were incubated with a primary antibody overnight at 4 °C. Primary antibodies obtained from Santa Cruz Biotechnology (Heidelberg, Germany) were iPla2β (cat# sc-14463), Xbp-1s (cat# sc-8015), Chop (cat# sc-7351), Srsf3 (serine/arginine-rich splicing factor 3) (cat# sc-13510), Fxr (cat# sc-25309), and calnexin (cat# sc-70481). Other primary antibodies were p-Ire1α (cat# NB100-2323, Novus Biologicals Europe, Abingdon, UK), Scd-1 (cat# ab19862, Abcam, Berlin, Germany), p-eIF2α (cat# 1090-1, Epitomics, Burlingame, CA, USA), Bip (cat# 3177, Cell Signaling, Frankfurt, Germany), p-Perk (cat# 3179, Cell Signaling), Gapdh (cat# 2118, Cell Signaling), and β-actin (cat# A1978, Sigma, Taufkirchen, Germany). After incubation with an HRP-conjugated secondary antibody, blots were developed by using a Luminata Forte ECL reagent (Millipore, Darmstadt, Germany).

### 2.4. Histology and Immunohistochemistry (IHC)

Liver and ileum specimens were fixed in formalin for at least 18 h and embedded in paraffin blocks, which were cut into 5-μm sections. Sections were stained with hematoxylin and eosin (H&E) or Sirus-Red according to standard protocols. For IHC, after deparaffinization and hydration, liver sections were subjected to antigen retrieval and subsequently treated with hydrogen peroxide to block endogenous peroxidase. Sections were exposed to a rabbit α-smooth muscle actin (α-SMA) antibody (cat# ab32575, Abcam) or anti-CK19 (cytokeratin 19) antibody (cat# ab133496, Abcam) overnight at 4 °C followed by a goat anti-rabbit secondary antibody (cat# ab6721, Abcam) for 1 h at room temperature. Positive staining was detected by diaminobenzidine, and slides were counterstained with hematoxylin. H&E-, Sirus-Red-, and IHC-stained cells were visualized with an Olympus IX 50 microscope. Quantification of positive staining was performed, using Image J (https:imagej.nih.gov/ij/download.html). For calculation of the length of the ileal villus, an image was taken at ×200 magnification corresponding to 0.6 mm^2^ of tissue. Number of stained positive area and length of ileal villus on each slide were counted from ten randomly selected fields. Evaluation of IHC sections was performed blindly.

### 2.5. Hepatic Profiling of Fatty Acids and PL

The levels of esterified and unesterified fatty acids (FA) contents in serum were determined by gas chromatography-mass spectrometry (GC-MS) under instrument conditions indicated in our previous studies [14,15]. For determination of PL in liver homogenates, a direct flow injection electrospray-ionization-tandem mass spectrometry (ESI-MS/MS) in positive mode was used under instrument conditions indicated in our previous studies [14,15]. For PL profiling of liver ER, isolated ER fractions at 300 μg proteins were used and subjected to Folch lipid extraction in the presence of internal standards, including 17:0 LPC, 14:0/14:0 PC, 12:0/12:0 PE, and 17:0 ceramides. Lipid extracts in methanol were analyzed by using a triple-quadrupole Micro Mass Quattro Premier MS/MS system coupled with LC equipment consisting of a binary and isocratic pump connected to an HTC Pal autosampler (CTC Analytics, Zwingen, Switzerland). Running conditions detection and data analysis of phospholipid subclasses have been described in our previous studies [16,20].

### 2.6. BA Profiling by LC-MS/MS and Liver Cholesterol

BA extraction was performed in the presence of an internal standard ursodeoxycholic acid-D_4_ according to our published method [20,26]. BA profiles were analyzed by an LC-MS/MS system, consisting of a separate module of a Waters 2695 and on-line detection by an electrospray ionization source of the tandem mass spectrometer (Quattro Micro API, Waters, UK), with previously described instrument conditions [20,26]. For determination of cholesterol, lipid extraction of the liver was performed, as previously described [16], and subjected to cholesterol measurement by enzymatic assay kits (Randox, Krefeld, Germany). 

### 2.7. Gene Expression by RT-PCR

Total liver RNA was prepared using GenElute™ Mammalian Total RNA Miniprep Kit (Sigma). RNA was reverse transcribed to cDNA using Maxima^®^ First-Strand cDNA synthesis Kit (Thermo Fisher Scientific, Dreieich, Germany). The quantitative real-time polymerase chain reaction was performed using Applied Biosystems (7500 Fast Real time PCR system) with TaqMan^®^ gene expression assays and TaqMan^®^ Universal PCR Master Mix. The expression level of a target gene was calculated using comparative Ct (∆∆ Ct) method and determined as a ratio of the target normalized to a house-keeping gene Gapdh.

### 2.8. Statistics

Results were expressed as mean ± SEM. *p* < 0.05 was considered significant by using pairwise Student’s *t*-tests or Kruskal-Wallis tests with Dunns’ selected pair posttests of GraphPad Prism7.

## 3. Results

### 3.1. iPla2β Deficiency during MCD Induced Liver Fibrosis without Affecting Steatosis

As female mice [27] and humans [28] have been reported to show an adverse response due to choline deficiency, we, therefore, subjected female WT and iPla2β^−/−^ mice to MCD feeding for 5 weeks [16]. MCD feeding caused a significant reduction of body weights and liver weights to the same extent in WT and iPla2β^−/−^ mice (Figure 1A). iPla2β^−/−^ mice fed with MCD showed an attenuation trend of alanine transferase (ALT) without altering serum glucose and esterified + unesterified fatty acids (Figure 1B). This deficiency also did not affect MCD-induced elevation of liver triglycerides (TG) and free fatty acids (FFA) (Figure 1C), indicating no effects on hepatic steatosis. As expected [11], MCD feeding of WT mice suppressed hepatic PC contents, and unlike our HFD study [15], this suppression was not altered by iPla2β deficiency (Figure 1D). Consistent with our previous results in male mice [14,15], female iPla2β^−/−^ mice also showed decreased levels of the products LPC and LPE (lysophosphatidylethanolamine) containing saturated fatty acids in livers after chow or MCD feeding. This indicates iPla2β specificity for the hydrolysis of PL containing saturated fatty acids at sn-1 position in the hepatocytes of female mice. 

Upon examining H&E stained livers, WT and iPla2β^−/−^ mice fed with MCD diet similarly developed severe hepatocellular ballooning and lipid-droplet accumulation in hepatocytes (Figure 1E). MCD feeding of WT mice did not markedly increase the positivity of Sirius-Red and α-SMA [16]; however, iPla2β^−/−^ mice fed with MCD showed a marked increase of these markers (Figure 1E).

Compared with MCD-fed WT mice, iPla2β^−/−^ mice fed with MCD showed a significant increase of α-SMA, TGF-β1, collagen3α1 (Figure 1F), and monocyte chemoattractant protein-1 (MCP-1) (Figure 1G) mRNA expression. Only a trend increase was observed for collagen4α1 and MCP-1 receptor CCR2 mRNA expression. Thus, these data confirmed our previous study [16] for the susceptibility of liver fibrosis in iPla2β^−/−^ mice fed with the MCD diet.

### 3.2. Effects of iPla2β Deficiency on Hepatic ER UPR in Mice Fed with Chow or MCD

As iPla2β is localized in the ER [21], we investigated molecular mechanisms at the levels of ER in female WT and iPla2β^−/−^ mice fed with chow or MCD for 5 weeks. We previously reported that the intestine of aged male iPla2β^−/−^ mice shows suppressed expression of total and spliced Xbp-1 (Xbp-1s) [20]. Xbp-1 is a downstream target of ER transmembrane protein inositol-requiring enzyme1α (IRE1α) recognized as a protective cellular signaling pathway in response to an accumulation of misfolded proteins [29]. However, BA, classified as FXR agonists [30], and saturation of membrane lipids [31] have been shown to also induce ER UPR independent of accumulation of misfolded proteins. Here, we showed that iPla2β^−/−^ mice fed with chow already exhibited suppressed hepatic expression of phosphorylated-Ire1α (p-Ire1α) and its downstream targets, i.e., Xbp-1s [29] and serine/arginine-rich splicing factor 3 (Srsf3) [32] proteins (Figure 2A). Compared with MCD-fed WT, MCD-fed iPla2β^−/−^ mice showed further suppression of hepatic Xbp-1s and Srsf3, but an increase of p-Ire1α expression. While only suppression trend of an Xbp-1 target stearoyl-CoA desaturase 1 (Scd1) was observed (Figure 2A), MCD feeding of WT mice markedly caused suppression of Srsf3 targets *Srebp-2* (for cholesterol synthesis) and *Srebp-1c* (for de novo fatty acid synthesis) (data not shown). These targets were, however, not altered by iPla2β deficiency.

ER UPR involves activation of three transmembrane proteins IRE1α, protein kinase RNA-like ER kinase (PERK), and activating transcription factor 6 (ATF6). While no significant changes of Atf6 were observed in mutant mice under chow or MCD (data not shown), the expression of phosphorylated-Perk (p-Perk) and its downstream target CCAAT-enhancer-binding protein homologous protein (Chop) was decreased in livers of mutant mice fed with chow or MCD (Figure 2B). MCD feeding of WT mice, on the other hand, activated downstream of PERK pathway by increasing expression of phosphorylated-eukaryotic translation initiation factor 2α (p-elF2α) and the chaperone binding immunoglobulin protein (BiP) (Figure 2B). Both of these targets were, however, not modulated by iPla2β deficiency. Taken together, iPla2β deficiency in chow-fed mice showed suppressed expression of ER UPR Xbp-1s, Srsf3, p-Perk, and Chop, and some of which were further downregulated by MCD feeding. Consistent with a previous report [19], MCD feeding of WT mice did not activate all ER stress markers. Rather, this feeding caused downregulation of Srsf3 and Chop, and iPla2β deficiency further downregulated Srsf3 expression.

### 3.3. Altered ER Membrane Phospholipids by iPla2β Deficiency in Mice Fed with Chow or MCD

We previously reported that iPla2β catalyzes the hydrolysis of PC and PE to LPC and lysophosphatidylethanolamine (LPE), respectively, in ER fractions isolated from livers of male mice [15]. Since activation of ER transmembrane p-Ire1α and p-Perk was decreased in chow-fed mutant mice (Figure 2A,B), we surmised that ER membrane PL could be altered by iPla2β deficiency or combined with MCD. Our ER preparations from livers led to an enrichment of a resident ER protein calnexin in ER fractions but not in liver homogenates (Figure 3A). An absence of iPla2β protein expression was observed in both liver homogenates and ER fractions (Figure 3A, left). In chow-fed WT mice, p-Ire1α expression was higher in ER fractions than that in liver homogenates (all samples with 10 μg proteins loaded, Figure 3A, right), indicating Ire1α localization in the ER. iPla2β deficiency decreased p-Ire1α expression in the ER fraction, which is consistent with the observation in liver homogenates (with 30 μg proteins loaded, Figure 2A).

PL profiles of liver ER fractions were analyzed as PL classified into those containing saturated, monounsaturated (MUFA), polyunsaturated (PUFA) fatty acids, and total PL (Figure 3B–H). Due to substrate depletion of PC synthesis [11], MCD feeding of WT caused a strong reduction of LPC (Figure 3B), PC (Figure 3C), and sphingomyelin (SM) (Figure 3D), but not LPE (Figure 3E) and PE (Figure 3F). Compared with MCD-fed WT, MCD-fed iPla2β^−/−^ mice showed a further decrease of saturated-and total LPC (Figure 3B), which led to a further increase of PC/LPC ratio (Figure 3I). This demonstrated iPla2β specificity for PC under MCD condition. Notably, chow-fed mutant mice showed a significant decrease of saturated-and total LPE, indicating iPla2β specificity for PE under normal chow (Figure 3E). Compared with MCD-fed WT, the total LPE levels were further decreased in MCD-fed mutant mice. As LPC and LPE are products of iPla2β under chronic HFD feeding [15], our results also demonstrated this activity under MCD feeding.

MCD feeding of WT mice did not markedly alter ER PE contents (Figure 3F), but significantly decreased not only PC (Figure 3C) and SM (Figure 3D) but also phosphatidylinositol (PI) (Figure 3G), phosphatidylserine (PS) (Figure 3H), particularly those containing PUFA. This indicated a defect of PL remodeling associated with MCD-induced hepatic steatosis. More importantly, MCD-fed iPla2β^−/−^ mice showed a further decrease in these PUFA-containing PL, indicating that there was an insufficient number of PUFA molecules for reacylation. Hence, MCD-induced defect of ER PL remodeling became more severe by iPla2β deficiency. 

It has been shown that a decrease of PC/PE ratio in the ER is associated with a decrease of p-Ire1α [33]. Consistently, iPla2β^−/−^ mice fed with chow showed a decrease of ER PC/PE ratio (Figure 3I) and p-Ire1α expression (Figure 3A) as well. This PC/PE decrease was a result of an increase of PE (Figure 3F) due to iPla2β inactivation and possibly activation of CoA-independent acyltransferase [16]. MCD-fed WT mice also showed a decrease of ER PC/PE ratio due to suppression of PC synthesis induced by MCD. This decrease was reversed by iPla2β deficiency due to a further decrease in PE (Figure 3F). As MCD feeding severely suppressed ER PI and PS levels (Figure 3G,H), hence PC/PS and PC/PI ratios were increased (Figure 3I). These ratios were even further increased by iPla2β deficiency, thus supporting more severe ER PL remodeling defect.

### 3.4. iPla2β Deficiency under MCD Altered Hepatic and Intestinal Fxr and BA Transport Genes

As Xbp-1 is shown to regulate the key nuclear receptor Fxr [30], we analyzed hepatic Fxr protein expression. Upon steatosis induction, MCD feeding of WT mice caused suppression of Fxr protein, which was further suppressed by iPla2β deficiency (Figure 4A). On the mRNA levels, MCD feeding of WT mice suppressed the expression of *Fxr*, its downstream nuclear receptor small heterodimer partner (*Shp*), and cholesterol 7α-hydroxylase (*Cyp7a1*), the rate-limiting enzyme of BA synthesis (Figure 4B) [34]. iPla2β deletion during MCD rescued the suppression of *Cyp7a1*, attenuated MCD-induced elevation of bile salt export pump (*Bsep*) (which exports BA to bile ducts), and further attenuated expression of multidrug resistance protein 3 (*Mrp3*) (which exports BA to the hepatic artery). The latter two effects were consistent with FXR function in the regulation of BA transporters [35]. 

We have shown that aged male iPla2β^−/−^ mice show suppressed expression of intestinal Xbp-1s and Fxr associated with intestinal atrophy [20]. Consistently, female iPla2β^−/−^ mice on a chow diet already showed some abnormal shrinkage of ileal villous length (Figure 5A). MCD feeding of WT mice also caused severe ileal villous shrinkage associated with exposed lamina propria, and this damage was further exaggerated in MCD-fed iPla2β^−/−^ mice showing disturbed mucous membrane and mucosal atrophy. Furthermore, it has been reported that FXR is a regulator of a negative feedback loop that controls expression of Cyp7a1 in the liver via intestinal induction and release of fibroblast growth factor 15 (FGF15) into a portal vein [34]. Similar to liver *Fxr* and *Shp* (Figure 4B), the ileum of iPla2β^−/−^ mice fed with MCD showed significant downregulation of *Fxr* and a decreasing trend for *Fgf15* (Figure 5B). Overall effects of iPla2β deficiency during MCD on intestinal BA transport genes were relatively insignificant and only seen as the rescue of suppressed *Abst* expression. We concluded that iPla2β inactivation or MCD alone had an intrinsic effect in suppressing *Fxr* in mouse liver (Figure 4) and ileum (Figure 5B). 

### 3.5. iPla2β Deficiency during MCD Exacerbated Muricholic Acid and Its Tauro-Conjugate in Liver and Peripheral Blood

Since BA transport genes were altered in MCD-fed iPla2β^−/−^ mice, we analyzed the profile of BA species in the liver, ileum, bile, portal vein serum, and *vena cava* serum by using LC-MS/MS (Figure 6A–E). MCD-fed WT mice did not show significant changes of BA in the liver (Figure 6A) and ileum (Figure 6B) but showed a significant increase of primary BA cholic acid (CA) and muricholic acid (MCA) in bile (Figure 6C). These BA, together with tauro-MCA (TMCA) and tauro-LCA (lithocholic acid) (TLCA), were increased in portal vein serum (Figure 6D). Moreover, TMCA and CA were increased in *vena cava* serum of MCD-fed WT mice (Figure 6E). The elevation of BA species in bile and blood by MCD feeding was consistent with a previous report [11]. Compared to MCD-fed WT, MCD-fed iPla2β^−/−^ mice showed an increase of MCA and TMCA in the liver (Figure 6A), and these BA together with tauro-cholic acid (TCA) were also increased in *vena cava* serum (Figure 6E). These increases of BA in *vena cava* serum were associated with the observed intestinal damage (Figure 5A), which would allow BA leakage into peripheral blood [20,26].

Total BA levels with all BA species combined were analyzed to compare BA in the enterohepatic circulation. MCD feeding of WT mice induced an increase of total BA seen only in *vena cava* serum (Figure 6F). iPla2β deficiency under MCD further increased total BA (i.e., MCA + TMCA, Figure 6A) in the liver and decreased total BA (i.e., TLCA, Figure 6D) in portal vein serum. Notably, iPla2β^−/−^ mice fed with chow also showed an increase of total BA in the portal vein and *vena cava* serum (Figure 6F) likely due to mild intestinal damage in these mice (Figure 5A).

Following the classification of BA species in the liver, MCD feeding of WT mice induced an increase of primary BA, BA classified as Fxr antagonists, and conjugated BA (Figure 6G). iPla2β deficiency during MCD caused a further elevation trend of Fxr antagonists [36], i.e., MCA + TMCA (Figure 6A). Associated with hepatic steatosis, MCD feeding of WT mice induced an elevation of liver cholesterol, which was attenuated by iPla2β deficiency (Figure 6H). This attenuation, together with an increased liver BA (Figure 6A), might suggest an increase of BA synthesis in livers of MCD-fed mutant mice. Thus, iPla2β deficiency under MCD caused an elevation of MCA + TMCA in liver and MCA + TMCA + TCA in *vena cava* serum.

Since BA can stimulate proliferation of cholangiocytes [37], and MCD-fed mutant mice showed increased liver BA, we, therefore, performed IHC of cytokeratin 19 (CK19). With no effects with MCD or iPla2β deficiency alone, MCD-fed mutant mice showed a marked increase of CK19 expression (Figure 7A). Hence, iPla2β deficiency rendered exacerbation of biliary epithelial cell proliferation during MCD-induced lean NASH.

## 4. Discussion

Impaired liver functions, such as fibrosis and cryptogenic cirrhosis, are seen in lean NASH patients [6,7,8], and BA metabolism plays a critical role in this disease [11,21,22,23,35]. We determined the mechanisms of iPla2β deficiency in MCD-fed female mice, which showed exaggerated liver fibrosis [16]. iPla2β deficiency combined with MCD further downregulated ER UPR proteins associated with altered liver ER PL composition with a severe defect of PL remodeling. These changes were associated with activation of biliary epithelial cell proliferation concomitant with an increase of MCA + TMCA in the liver and peripheral blood due to altered expression of BA transport genes and intestinal damage, respectively (Figure 7B). iPla2β deficiency alone had a marked suppressive effect on homeostatic genes, i.e., Fxr in liver and intestine, liver ER UPR proteins, and an elevated BA in the blood (Figure 7C). Hence, our work provides new insights into mechanisms of iPla2β inactivation on ER PL, UPR, and Fxr, which resulted in increased BA under normal conditions, as well as liver fibrosis [16] and biliary epithelial cell proliferation during MCD-induced lean NASH.

Recent work in our laboratory has revealed contrasting phenotypes of iPla2β^−/−^ mice, on the one hand, showing susceptibility during aging [20] and autoimmune liver injury [26], but, on the other hand, showing protection against genetic [14] and HFD [15]-induced obesity and hepatic steatosis. The mechanism for this protection involved the replenishment of PUFA-containing PL and correction of PL remodeling defect. Associated with no hepatic steatosis protection, iPla2β inactivation during MCD was unable to correct PL remodeling defect as observed in liver homogenates [16] and liver ER (Figure 3), and rather, more severe PL remodeling defect was observed in liver ER (Figure 7B). We showed that the pathogenesis of MCD-induced fatty liver did not involve ER stress, which is in line with a previous report [19]. Rather than ER stress activation, MCD feeding caused inactivation of ER UPR proteins, Fxr, and a defect of ER PL remodeling. MCD combined with iPla2β deficiency further suppressed Fxr, Xbp-1s, Srsf3, p-Perk, and Chop expression associated with a more severe PL remodeling defect (Figure 7B). This suppression of homeostatic genes was concomitant with liver fibrosis [16] and activated biliary epithelial cell proliferation considering a biliary repair from fibrosis [38] and cholestasis [39].

Recent studies have revealed that the ER UPR system is sensitive to the disequilibrium of ER lipids [40,41]. Independently of changes to protein folding homeostasis in the ER lumen, studies in yeast have shown that ER UPR can be activated by changes of different lipids, such as lipid saturation [31], sterol, sphingolipids, PC/PE ratio, inositol, and membrane PC desaturation [40,41,42,43,44]. This UPR activation is based on biophysical principles in the regulation of transmembrane protein dimerization [40,41]. While an increase of ER PC/PE ratio is reported in *ob/ob* [18] and HFD-fed [15] mice, a decrease of this ratio, on the other hand, has been observed in MCD-fed WT mice and another lean NASH model [10,45]. This difference is consistent with a U-shape curve of liver PC/PE ratio among various NAFLD/NASH models [45]. This also suggests certain flexibility in alterations among PL subclasses to describe NAFLD/NASH pathogenesis. In line with this, studies have shown that the remodeling of PUFA-containing PI also plays a role in NAFLD [46]. We showed here that MCD combined with iPla2β deficiency caused multiple aberrancies of ER PL subclasses characterized by an increase of PC/LPC, PC/PE, PC/PS, and PC/PI ratios (Figure 7B). The ability of iPla2β inactivation to decrease PE, PS, and PI supports iPla2β role on the hydrolysis of PUFA-containing PL. By biophysical distortions of the ER membrane, we surmise that changes in ER LPC, PE, PS, and PI relative to PC altogether may prevent the dimerization of Ire1 and Perk [40,41] and/or induce inactivation of insertase, thus preventing protein insertion into ER membrane [47]. The observed decrease of ER LPC levels would likely promote a loss of ER membrane positive curvature [48]. Together with membrane curvature loss [49,50], the decrease of positively charged PL, such as PE and PS, may prevent protein recruitment via amphipathic helix and protein insertion to the ER membrane. As Ire-1α uses an amphipathic helix to sense membrane aberrancies [40], changes in four PL, namely LPC, PE, PS, and PI, relative to PC may lead to an inhibition of its insertion, leading to decreased contents and activation of Ire-1α (and Perk) in ER membrane. Further experiments are warranted to test this complex mechanism. Nonetheless, the downregulation of Ire-1α in the ER would lead to the suppression of Xbp-1s, Srsf3, and Chop proteins by iPla2β deficiency combined with MCD. Similar to iPla2β [21] (Figure 3A), an Ire-1α -target Srsf3 is also localized in the ER and Golgi [51]. Srsf3 has multiple signaling pathways important for hepatocyte homeostasis, and its deletion renders susceptibility to liver injury [51]. The suppressed Srsf3 expression in MCD-fed iPla2β^−/−^ mice was in line with observed exaggeration of liver fibrosis in these mice [16] (Figure 1 and Figure 7B).

As ER UPR is linked to Fxr [30], MCD alone or combined with iPla2β deficiency consistently suppressed hepatic Fxr expression. Fxr is an important homeostatic gene in BA homeostasis, and its suppression is reported to downregulate BA transporter expression [35] and increase BA levels in liver and serum [52]. This severe suppression of Fxr may also lead to exaggerated activation of hepatic α-SMA [53], also observed in MCD-fed mutant mice (Figure 1F) [16]. Indeed, increased hepatic BA production is associated with increased NAFLD fibrosis score in patients [24]. This highlights iPla2β role in BA homeostasis via Fxr observed not only in liver injury models [20,26] but also in MCD-induced NASH (Figure 7B).

As XBP-1, FXR, and BA metabolism are closely linked [22,30], an accumulation of MCA + TMCA was consistently observed in livers and peripheral blood of MCD-fed iPla2β^−/−^ mice. This could be a result of the suppressed expression of hepatic Bsep, which is shown to be correlated with NAFLD progression [54]. Since MCA and TMCA are produced in mouse liver and secreted to the intestine, the observed elevation of MCA + TMCA in peripheral blood was likely due to severe intestinal damage in MCD-fed mutant mice. Furthermore, an accumulation of MCA + TMCA may lead to activation of hepatic stellate cells [55], which may contribute to the generation of progenitor cells, such as cholangiocytes [56]. Consistently, CK-19 staining of biliary epithelial cells was increased in MCD-fed iPla2β^−/−^ mice. This induction of a ductular reaction might lead to the initiation and progression of cholangiopathies in these mutant mice undergoing lean NASH (Figure 7B). As MCD induces inflammatory lean NASH by activation of Kupffer cells and monocyte-derived macrophages [57,58], the observed phenotypes proposed could be due to iPla2β deficiency in macrophages. Further studies are warranted to test whether macrophage-specific iPla2β^−/−^ mice would still exhibit the observed phenotypes following MCD feeding. 

MCD-induced NASH in mice showed a decrease of blood FFA (Figure 1B) as a result of an increase of FA uptake [59], thus resulting in hepatic steatosis. These mice also showed lower blood glucose (Figure 1B) and did not exhibit insulin resistance [17]. As lean subjects with evidence of NAFLD have clinically relevant impaired glucose tolerance [9], MCD-fed mice without hyperlipidemia and insulin resistance do not exhibit a full spectrum of human lean NASH. Our results for fibrosis susceptibility in MCD-fed iPla2β^−/−^ mice should, therefore, be interpreted with cautions regarding the pathogenesis of lean NASH. Since nutrient choline is a human dietary requirement, iPla2β inactivation under choline deficiency, on the other hand, may apply to those individuals who consume choline-deficient diet [28,60].

Consistent with the intestine of aged male iPla2β^−/−^ mice [20], chow-fed female iPla2β^−/−^ mice in our study already showed suppressed expression of ER UPR p-Ire1α, Xbp-1s, Srsf3, p-Perk, and Chop (Figure 7C). These mutant mice also showed suppressed Fxr in liver and intestine. The latter could lead to intestinal BA accumulation, as seen in peripheral blood, as a result of intestinal damage (Figure 7C). We found that iPla2β under normal conditions was specific for the hydrolysis of PE, and that iPla2β inactivation increased PE/LPE but decreased PC/PE ratio (Figure 7C). While these ratios are different from those seen in MCD conditions, the decrease of LPE might cause an alteration in curvature [48], and an increase of PE might contribute to alteration in charged PL of the ER membrane [49]. These parallel PL changes could lead to an inactivation of Ire1α and Perk by preventing oligomerization [40,41,42,43,44] and/or inhibiting the action of insertase for protein insertion into ER membrane [47]. Similar to cytosolic phospholipase A2 [48], iPla2β’s ability to hydrolyze PE and PC under normal and MCD conditions, respectively, appears to be a key mechanism contributing to membrane curvature changes of the ER membrane, leading to suppressed UPR. Stemming from altered ER membrane lipids due to iPla2β deficiency, iPla2β^−/−^ mice are susceptible to autoimmune hepatitis [26], and aged mutant show several abnormalities, including liver fibrosis [20], male fertility, bone mass density [12], and neurological disorders [61]. Consistently, homozygous mutations in the iPla2β or PLA2G6 gene have been reported in patients with dystonia-parkinsonism [62] and neuroaxonal dystrophy [63].

Taken together, genetic loss of iPla2β did not protect mice from MCD-induced hepatic steatosis but rather led to exaggerated hepatic fibrosis and biliary epithelial cell proliferation. This phenotype was associated with a severe defect in the remodeling of ER PL, severe suppression of ER UPR and Fxr, and hepatic BA accumulation. Taken together, iPla2β hydrolysis of ER PL was critical for regulation of homeostatic genes under normal conditions and during MCD-induced lean NASH. Our results may have implications on individuals with hepatic fibrosis and cholangiopathies with PLA2G6 mutations who exhibit NASH or consume choline-deficient diet [28,60].

## Figures and Tables

**Figure 1 cells-08-00879-f001:**
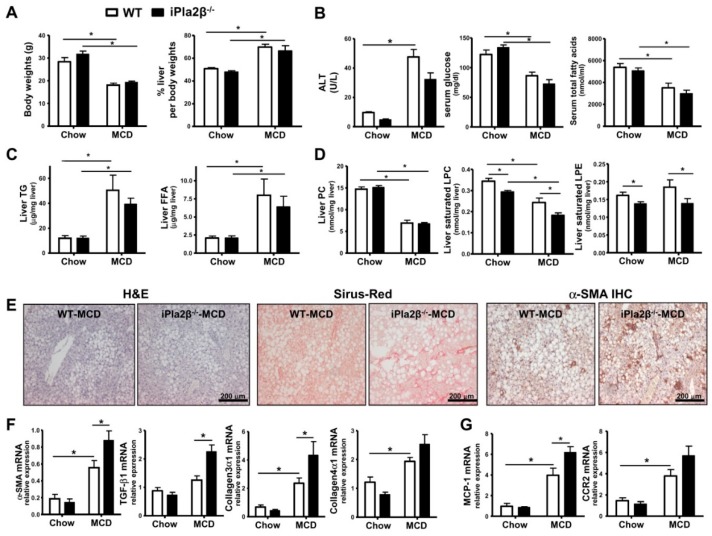
iPla2β (calcium-independent phospholipase A2) deficiency during MCD (methionine- and choline-deficient) induces liver fibrosis without affecting steatosis. Female WT (wild-type) and iPla2β^−/−^ mice were fed with chow or MCD diet for 5 weeks. (**A**) Body and % liver weights. (**B**) Serum ALT, glucose, and total fatty acids determined by GC-MS. (**C**) Liver triglycerides (TG) and non-esterified or free fatty acids (FFA). (**D**) Liver PC (phosphatidylcholine), saturated LPC (lysophosphatidylcholine), and saturated LPE (lysophosphatidylethanolamine) determined by LC-MS/MS. (**E**) Liver slides were subjected to H&E (hematoxylin and eosin), Sirius-Red, and α-SMA IHC (immunohistochemistry) staining. (**F**) Expression of α-SMA, TGF-β1 (transforming growth factor-β1), collagen3α1, and collagen4α1 mRNA. (**G**) Expression of MCP-1 (monocyte chemoattractant protein-1) and CCR2 (CC motif chemokine receptor 2) mRNA. Data are ± SEM, N = 4–7; * *p* < 0.05 between indicated groups.

**Figure 2 cells-08-00879-f002:**
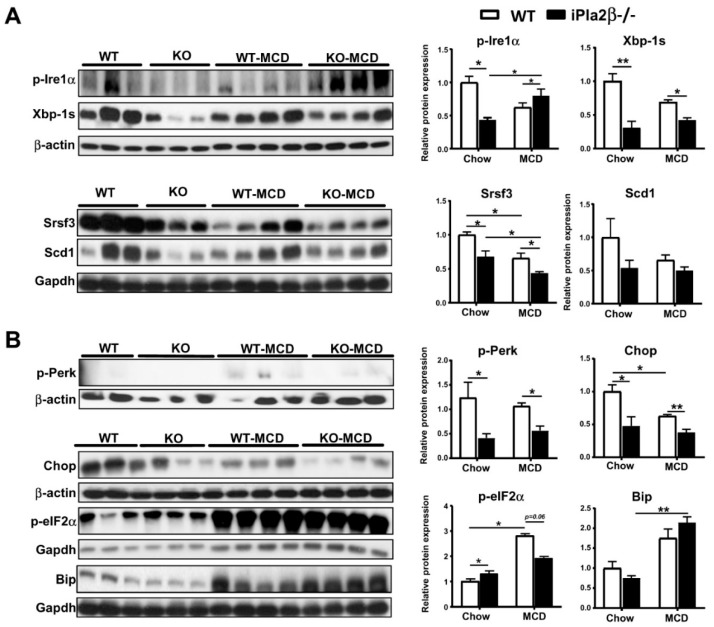
iPla2β deficiency and MCD causes suppression of hepatic adaptive UPR (unfolded protein response) proteins. Female WT and iPla2β^−/−^ mice were fed with chow or MCD diet for 5 weeks. Western blot analysis was performed with 30 μg proteins in each lane. (**A**) Hepatic protein expression of p-Ire1α (phosphorylated inositol-requiring enzyme1α), Xbp-1s (X-box binding protein-1s), Srsf3 (serine/arginine-rich splicing factor 3), and Scd1 (stearoyl-CoA desaturase 1) (left) and quantification (right). (**B**) Hepatic protein expression of p-Perk, Chop, p-eIF2α, and Bip (left) and quantification (right). Data are ± SEM, N = 5–7; * *p* < 0.05; ** *p* < 0.01 between indicated groups.

**Figure 3 cells-08-00879-f003:**
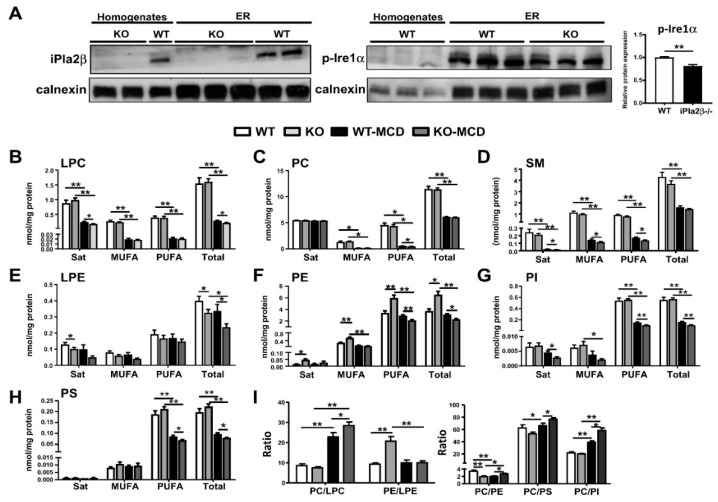
iPla2β deficiency alone or combined with MCD modulates the contents and composition of liver ER (endoplasmic reticulum) PL (phospholipid). Female WT and iPla2β^−/−^ mice were fed with chow or MCD diet for 5 weeks. ER fractions were isolated from mouse livers, and 300 μg ER proteins subjected to PL profiling by LC-MS/MS. (**A**) Protein activation of ER transmembrane expression of iPla2β (left) and p-Ire1α and quantification (right) in liver homogenates and ER fractions with 10 μg proteins in each lane. For PL profiling in liver ER fractions, saturated, monounsaturated (MUFA), polyunsaturated (PUFA) fatty acids, and total contents of (**B**) LPC, (**C**) PC, (**D**) SM (sphingomyelin), (**E**) LPE, (**F**) PE, (**G**) PI (phosphatidylinositol), and (**H**) PS (phosphatidylserine) were determined as nmol/mg protein. (**I**) PC/LPC, PE/LPE, PC/PE, PC/PS, and PC/PI ratios were determined from (**A**) to (**H**) data. Data are mean ± SEM, N = 6; * *p* < 0.05; ** *p* < 0.01 between indicated groups.

**Figure 4 cells-08-00879-f004:**
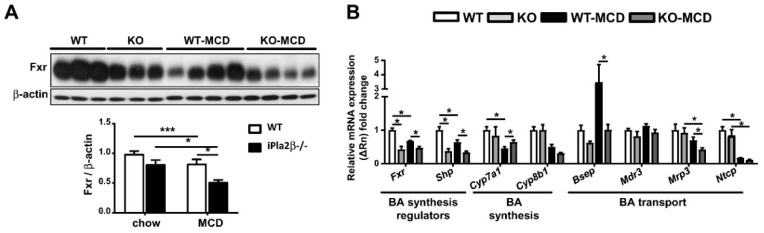
iPla2β deficiency alone or combined with MCD suppresses hepatic Fxr protein and alters mRNA expression of BA synthesis and transport genes. Female WT and iPla2β^−/−^ mice were fed with chow or MCD diet for 5 weeks. (**A**) Hepatic Fxr protein expression and quantification. Western blot analysis was performed with 30 μg proteins in each lane. (**B**) Hepatic mRNA expression of *Fxr*, *Shp*, *Cyp7a1*, *Cyp8b1*, *Bsep*, *Mdr3*, *Mrp3*, and *Ntcp*. Data are mean ± SEM, N = 5–7; * *p* < 0.05; *** *p* < 0.001 between indicated groups.

**Figure 5 cells-08-00879-f005:**
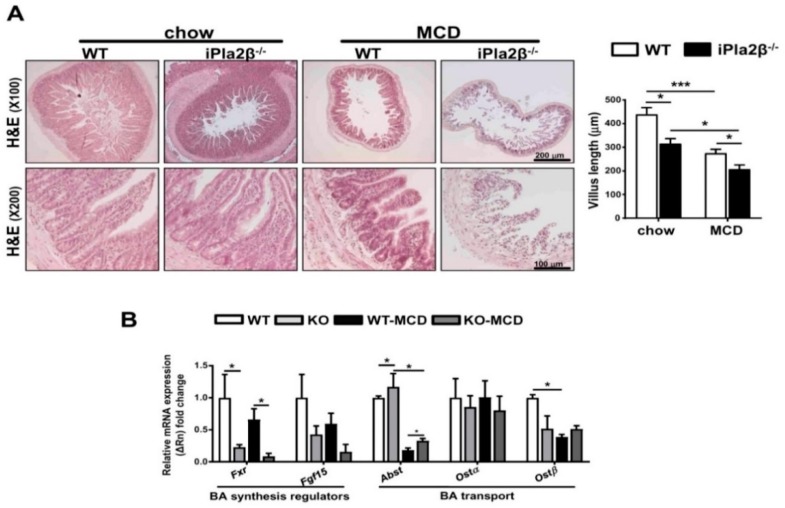
iPla2β deficient mice fed with MCD show enhanced intestinal villus atrophy and suppressed Fxr (farnesoid X-activated receptor) expression. Female WT and iPla2β^−/−^ mice were fed with chow or MCD diet for 5 weeks. (**A**) Representative histological H&E of the intestine (left) and quantification of ileal villus length (right). (**B**) Ileal mRNA expression of *Fxr*, *Fgf15*, *Abst*, *Ost*α, and *Ost*β. Data are mean ± SEM, N = 5–7; * *p* < 0.05; *** *p* < 0.001 between indicated groups.

**Figure 6 cells-08-00879-f006:**
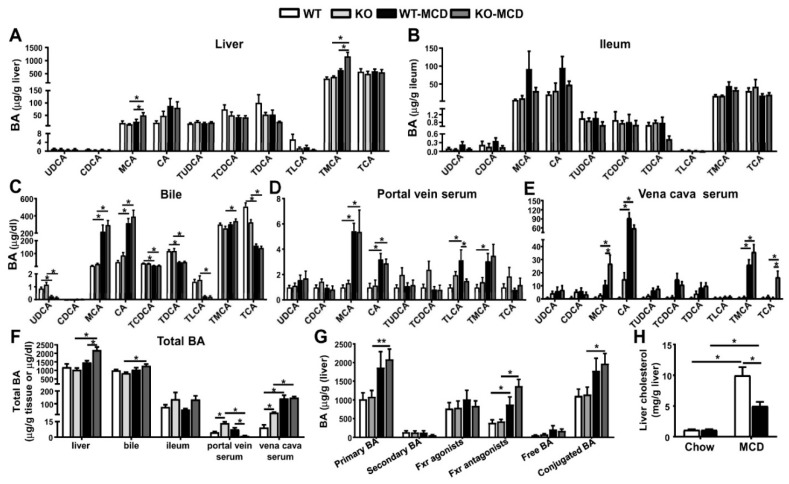
Effects of iPla2β deficiency during MCD feeding on BA (bile acids) contents in liver, ileum, and enterohepatic circulation. Female WT and iPla2β^−/−^ mice were fed with chow or MCD diet for 5 weeks. BA profiles of samples were determined by LC-MS/MS. BA profiles in (**A**) liver, (**B**) ileum, (**C**) bile, (**D**) portal vein serum, and (**E**) *vena cava* serum. (**F**) Total BA contents in liver, bile, ileum, portal vein serum, and *vena cava* serum. (**G**) BA classified as primary BA, secondary BA, Fxr agonists, Fxr antagonists, free BA, and conjugated BA. (**H**) Liver cholesterol levels. Data are mean ± SEM, N = 5–7; * *p* < 0.05; ** *p* < 0.01 between indicated groups.

**Figure 7 cells-08-00879-f007:**
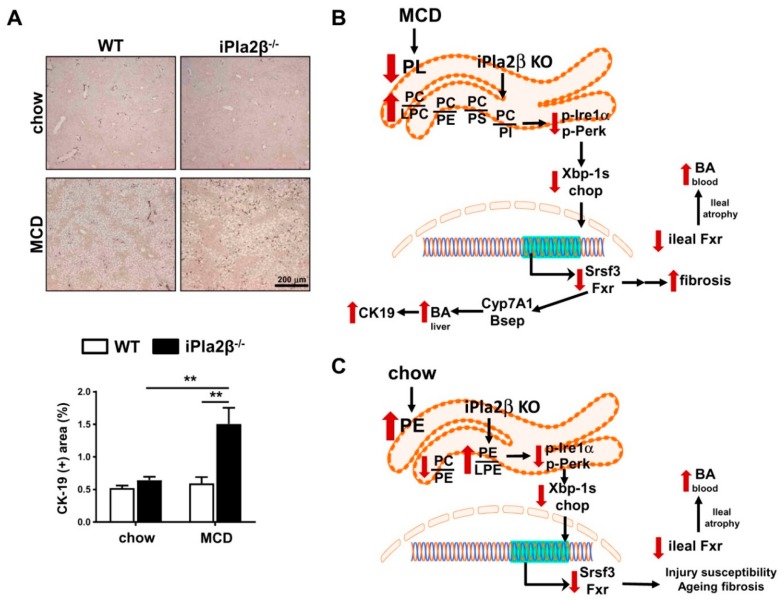
Effects of iPla2β deficiency during MCD feeding on biliary epithelial cells. (**A**) Livers of WT and iPla2β^−/−^ mice fed with chow or MCD diet for 5 weeks were subjected to IHC staining of biliary epithelial cell marker cytokeratin 19 (CK19) (top) and quantification of CK19 IHC-positive area (bottom, data are mean ± SEM, N = 5–7; ** *p* < 0.01 between indicated groups). (**B**) Proposed mechanisms of iPla2β deficiency during MCD-induced lean NASH (non-alcoholic steatohepatitis) showing altered ER PL composition leading to suppression of UPR Xbp-1, Srsf3, and Fxr, resulting in increased liver BA, liver fibrosis, and biliary epithelial cell proliferation. iPla2β^−/−^ mice fed with MCD showed fibroductular response associated with an increased BA in liver and blood. (**C**) Proposed mechanisms of iPla2β deficiency under normal conditions, showing altered ER PL, UPR Xbp-1, Srsf3, and Fxr. This might lead to previously reported susceptibility to liver injury and liver fibrosis during aging. Intestinal leakage of BA led to an increase of BA in peripheral blood in iPla2β^−/−^ mice fed with either chow or MCD.

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
