# Peer review of "iPla2β Deficiency Suppresses Hepatic ER UPR, Fxr, and Phospholipids in Mice Fed with MCD Diet, Resulting in Exacerbated Hepatic Bile Acids and Biliary Cell Proliferation"

_cells, 2019, doi:10.3390/cells8080879_

Round 1

Reviewer 1 Report

This work by Yanan et al. reports the iPla2β Deficiency Suppresses Hepatic ER UPR, Fxr, and Phospholipids in Mice fed with MCD Diet Resulting in Exacerbated Hepatic Bile Acids and Biliary Cell Proliferation. However, unfortunately, this work has many flaws in their experimental designs and thus their conclusions are not readily accepted. Most importantly, I don't think there is a successful induction of NASH in this study.  I suggest that authors rewrite it and also seek for an English editing.   Hence, I must reject this manuscript as current status of this manuscript.

Critiques are itemized as follows.

1. I understand that authors obtained detail data at molecular level for the calcium-independent phospholipaseA2 (iPLA2β) regulates homeostasis and remodeling of phospholipids (PL), but it cannot cover the less originality of this work and lack of direct evidence data. For example: in the histopathological examination lack of hepatic fibrosis marker data (fibronectin and α-SMA; I also don't see the phenomenon of liver fibrosis), serum biochemical measurement (ALT and AST levels), and tissue biochemical measurements were required. The authors should can refer to Jung et al., 2014,
Diabetes Research and Clinical Practice, 105, 47-57. and Cho et al., 2014, Dig Dis Sci, 59(7), 1416-1474.

2. I am faintly seeing iPla2β expression in Figure 2A. Please explain.

Author Response

Response to reviewer #1

Thank you for your comments. We had     an English speaker to help editing our manuscript.

To demonstrate successful induction of NASH, we have added new data in Figure 1 showing a     significant increase in MCP-1 in livers of MCD-fed WT mice. A 5-fold increase of TNFa mRNA expression was also reported in our     previous study in ref. 16. In new     Figure 1, we have shown characteristics of WT and mutant mice fed with     chow and MCD diet including body and liver weights, serum ALT, glucose,     fatty acids, liver TG, FFA. Evidence of liver fibrosis in MCD mutant     compared with WT mice is now included as sirus-Red, a-SMA IHC and mRNA expression of a-SMA,     TGF-b1, collagen3a1. These markers and MCP1 mRNA expression had not been published in     ref. 16. We have     scanned this Western blot film for better clear band of iPla2b, shown now     in new Figure 3. 

Reviewer 2 Report

In the current study authors describe the effects on hepatic endoplasmic reticulum (ER), ER unfolded protein response (UPR), ER membrane phospholipids, hepatic and intestinal Fxr and bile acids (BA) transport genes and BA species in 12 months old iPla2β -/- mice fed with standard chow or methionine-choline-deficient diet (MCD) for 5 weeks.

The aim of the authors according to abstract and discussion is to investigate the mechanisms of the exaggerated liver fibrosis previously observed in iPLA2β -/- mice that were fed with methionine-choline-deficient diet (MCD). However, there is any data in the current study showing the amount of liver fibrosis of the animals. Also other fibrosis factors like α-SMA or factors directly involved in the extracellular matrix are not measured. This difficult to associate the observed changes with the fibrosis process.

Also authors describe the MCD model as representative of lean human patients with NASH. However, limitations of MCD model as a NASH model should be discussed. In this regard, the  observed changes in phospholipids or BA may be highly dependent of the model (deficient diet) and, therefore, translation to humans should be considered even more cautiously.

Other comments:

Introduction. Line 61. “ Further investigation  (…)  lean NASH considered more inflammatory than obese NAFLD”. A reference should be added.

Results. Results of body weight and liver weight on the 4 groups of animals may be included.

Resullts. Line 197: “Since activation of ER transmembrane p-Ire1α and p-Perk was decreased,…”. Specifify the groups that are being compared.

Fig 2A. Seems that only results from standard chow are presented. Why are results of MCDD mice not shown?

Fig 2A (right). Why is p-Ire1α result not shown for KO homogenates?

Fig 3A. Quantification of steatosis may be included.

Results. Line 306: “Fig 2D” should be “Fig 5D”

Results. Line 322-325 Legend of Fig 6A should be “(top)” and “(bottom)” instead of “(left)” and “(right)”

Discussion. Line 427. Exaggerated hepatic fibrosis could not be concluded from your results

Author Response

Response to reviewer #2

Now, the evidence for liver fibrosis in MCD mutant mice has been shown in Figure 1.

Thank you for the insights. We have added a paragraph in the Discussion on page 14 regarding limitation of MCD mouse model. We also added a reference concerning liver dysfunctions in individuals consuming choline deficient diet which may be applicable in our study.

In Introduction, ref. 17 is the citation showing that MCD-induced NASH is more inflammatory than HFD-induced NAFLD.

We have added body weight and liver weight data in new Figure 1.

We have indicated on Line 197: “Since activation of ER transmembrane p-Ire1α and p-Perk was decreased,…” as in chow-fed mutant mice” on page 8.

In new Fig. 3A, we did not perform ER isolation of livers of WT and mutant mice fed with MCD because of insufficient amount liver tissues for the isolation. We had to split liver tissues into several assays and substantial amounts were sent to Dr. Liebisch for LC/MS-MS and GC-MS analyses. The results in Fig. 2A are intended to demonstrate iPla2b abundance in liver homogenates and the ER.

 Not only new Fig 3A (right), p-Ire1α expression in WT and KO liver homogenates has been already shown in new Fig. 2A.

 As reported in ref. 16, the quantification of steatosis and liver TG now shown in new Figure 1 now shows no difference among WT and mutant mice fed with MCD.

Correction is made in Results, Fig. 2D is now Fig. 6D. Thank you.

Correction is made in Figure legend of new Figure 7. Thank you. 

With our new data in Figure 1, exaggerated hepatic fibrosis could be concluded in the discussion.

Round 2

Reviewer 2 Report

Authors have provided fibrosis data that support the conclusions in the revised version of the manuscript,